# Biocompatibility of Plasma-Treated Polymeric Implants

**DOI:** 10.3390/ma12020240

**Published:** 2019-01-12

**Authors:** Nina Recek

**Affiliations:** Department of Surface Engineering and Optoelectronics, Jožef Stefan Institute, Jamova cesta 39, 1000 Ljubljana, Slovenia; nina.recek@ijs.si; Tel.: +386-1-477-36-72

**Keywords:** biomaterial, polymer, plasma, functionalization, surface properties, thrombosis, hemocompatibility, endothealization, vascular graft, biocompatibility, endothelial cells

## Abstract

Cardiovascular diseases are one of the main causes of mortality in the modern world. Scientist all around the world are trying to improve medical treatment, but the success of the treatment significantly depends on the stage of disease progression. In the last phase of disease, the treatment is possible only by implantation of artificial graft. Most commonly used materials for artificial grafts are polymer materials. Despite different industrial procedures for graft fabrication, their properties are still not optimal. Grafts with small diameters (<6 mm) are the most problematic, because the platelets are more likely to re-adhere. This causes thrombus formation. Recent findings indicate that platelet adhesion is primarily influenced by blood plasma proteins that adsorb to the surface immediately after contact of a synthetic material with blood. Fibrinogen is a key blood protein responsible for the mechanisms of activation, adhesion and aggregation of platelets. Plasma treatment is considered as one of the promising methods for improving hemocompatibility of synthetic materials. Another method is endothelialization of materials with Human Umbilical Vein Endothelial cells, thus forming a uniform layer of endothelial cells on the surface. Extensive literature review led to the conclusion that in this area, despite numerous studies there are no available standardized methods for testing the hemocompatibility of biomaterials. In this review paper, the most promising methods to gain biocompatibility of synthetic materials are reported; several hypotheses to explain the improvement in hemocompatibility of plasma treated polymer surfaces are proposed.

## 1. Introduction

In the developed world, cardiovascular diseases are the most frequent cause of morbidity and mortality of the population, and represent one of the greatest health problems. In Europe alone, the cost of treating patients with these diseases is over 200 billion euros a year. In the first place is atherosclerosis, which causes the internal walls of the vessels to constrict, which means the blood can no longer run freely through the veins, and therefore its flow slows down. Treatment of such diseases is possible with a vascular stent, or by replacing a damaged vessel with a synthetic vascular implant. Approximately 500 surgeries per year are performed per million inhabitants, in which the damaged vein is replaced by a vascular implant (artificial blood vessel). This number is still growing every year. Both treatment options are commonly used, but in the long term, the recovery of patients with vascular stent and, in particular, artificial blood vessel, is still unsatisfactory. About 10% of patients with artificial vessels experience post-operative complications, mainly due to inflammatory reactions, infections and aneurysms. In such cases, it is necessary to replace the artificial vessel with an autologous vein, which further increases the cost of treatment [1].

Therefore, for the treatment of highly calcified vascular constrictions, a surgical procedure is necessary, where by inserting a synthetic vascular implant, a bypass to restore the blood flow is made. The materials used in cardiovascular applications for prosthetic heart valves, catheters, heart assist devices, hemodialysers, synthetic vascular implants and stents have to meet the requirements for biocompatibility/hemocompatibility and should also have appropriate mechanical properties, in particular the flexibility and ease of surgical implantation [2,3]. Today, the following polymers are used for this purpose; polyamids, polyolefin, polyesters, polyuretans, polyethylene terephthalate and polytetrafluoroethylene [4]. All these materials have been used for synthetic vascular prosthesis for many years, but, unfortunately, they do not offer sufficient hemocompatibility, especially when used for replacement of veins of smaller diameters (<6 mm). The main reason for this is that the probability of thrombosis occurrence is even greater in the narrower part of the veins. On the wall of the artificial vein, there is a non-specific binding of plasma proteins, which also affects the platelet binding and is one of the main causes of thrombosis [5]. Lack of endothealization is another main cause of thrombosis. 

Biological response to biomaterials is very complex and still poorly known. Since the surface of the biomaterials is the one that enables the first interaction with the body, the properties of the surface of the biomaterials are of key importance for an appropriate biological response. For years, the most suitable materials were inert materials that do not react with the body and do not allow the integration of biomaterials with the body. Today, the opinion is that biocompatible materials that are in contact with blood should enable interaction with the body and prevent infections, inflammatory reactions, blood clotting and other related reactions. For hemocompatible materials, it is particularly important that the surface has anti-thrombogenic properties that prevent the occurrence of thrombosis. Thrombosis begins with the binding of plasma proteins to the surface of the biomaterial and is strongly dependent on the physical and chemical properties of the surface of the biomaterial. Clinical studies showed that poly-l-lactic acid (PLLA) stent, which was the first absorbable stent implanted in humans, had low complication rates for thrombosis and thus very high hemocompatibilty. Another clinical study also showed that metallic base stent, coated with poly-d,l-lactide, used to carry the antiproliferative drug everolimus, lacked stent thrombosis and even ensured total vascular function restoration [6,7]. However, stents used in elder patients resulted in significantly higher target vessel failure rates compared with younger patients. Moreover, with increasing age, revascularization rates were also higher. In addition, there was no difference in stent thrombosis [8]. In the past two decades, the experimental and clinical studies have grown significantly, but there is a still need to develop materials that mimic the properties of natural cardiac tissues, i.e., composite materials. Furthermore, novel surface modifications should also be evolved to develop better biocompatible cardiac biomaterials.

In order to improve the properties of materials that are in contact with blood, various methods of surface treatment are used. In general, these methods are divided into mechanical and chemical. Mechanical methods of treatment are not particularly interesting because they often cause damage and changes in the other properties of the material that we want to avoid. Chemical processing methods are further divided into wet chemical methods, which include treatments with various chemical reagents in aqueous or other liquid media, and gaseous treatments, including plasma treatments, ion beam treatments, electron jets, photon jets (lasers), X-ray and other energy rays. To improve biocompatible properties, antithrombotic deposits are often used, such as heparin, albumin, or chitosan. In addition to this kind of application, the pre-treatment of artificial vessels with attachment of endothelial cells is also used to improve the properties. Covering the prosthetic implants in vitro with endothelial cells was first suggested by Heering et al. [9], although many polymer surfaces are not optimal for cell adhesion as such, unless modified. Nevertheless, all these methods have a limited degree of success [10,11,12,13]. One of the methods for improving the biocompatibility of materials is the use of gaseous plasma, which has many advantages over other methods. Before processing, samples do not need any special pre-treatment; procedures are quick and environmentally friendly. Plasma can also modify surface charge, roughness and polymer crystallinity, which has an important influence on cell adhesion [14,15]. For surface modifications, many different types of plasma can be applied, depending on requirements and the particular application. Depending on the gas used (e.g., oxygen, nitrogen, CF_4_), specific functional groups are formed on the surface and it can be either hydrophobic or hydrophilic [16,17,18,19].

## 2. Biomaterials

Biomaterials are materials that are either natural or synthetic and are used to regulate, supplement or replace the function of a tissue in the human body [20]. Their role is to replace or restore the function of an injured or degenerate tissue or organ. They are helpful to treat, improve performance or correct abnormalities, which all improve the quality of the patient’s life. Biomaterials used in the manufacture of medical devices are metals, ceramics, composites and polymers. Metals are relatively strong, flexible and fairly resistant to wear. Their disadvantage is poor biocompatibility, corrosion and excessive strength compared to tissue and elimination of metal ions, which can eventually lead to allergic reactions. Ceramics are more compatible than metals and are more resistant to corrosion. Poor properties are fragility, demanding production and low mechanical durability and flexibility. Polymers are the most useful materials, since they are easy to manufacture and are available in various compositions and forms, such as, for example, gels, fibers, films, and solids. Polymers are widely used in many industries such as electronics, the automotive industry, the food industry, and nowadays their usefulness has gained great importance in the field of medical science. Polymers most commonly used for these purposes are polyurethane, silicone, polytetrafluoroethylene (PTFE), polyethylene (PE), polymethyl methacrylate (PMMA), polyethylene terephthalate, etc. The production of these materials is usually relatively simple, quick and cost effective. In addition, their physical and chemical properties are usually good, however, they are often too flexible and too weak to meet mechanical requirements for certain applications [21,22].

### 2.1. Artificial Vascular Grafts

In parallel with advances in vascular surgery, development and production of vascular implants were also carried out. Various substitutes are available for the replacement of a damaged or unusable blood vessel, which may either be biological or synthetic. Natural blood vessels inside the human body are arterial or venous and can be categorized as autologous, allograft and xenographic. Becouse of these differences in the size and anatomy of blood vessels it is not certain they will match between the donor and recipient host. Synthetic vascular implants are used as arterial supplements. The industry strives to produce materials with properties that are identical or, to the extent possible, similar to the real veins. In the synthesis, it is therefore necessary to consider certain criteria. The artificial vessels produced must be sterile and must not contain toxic substances. In addition, production processes must comply with strict regulations in this field. Production must be cost-effective. The artificial vessels must be flexible and elastic, but with time they must not lose their flexibility. During the post-implantation period, the expansion of the vessel must not exceed 15% of the internal diameter. Within five years of transplantation, a frequency of 2% for anastomotic aneurysms should be acceptable. In the same period, the infections were reported to occur in only 3% of cases. Depending on the size of the artificial vein, they are divided into veins with a large diameter (d > 6 mm), with a middle diameter (d = 4–6 mm) and veins with small diameters (d < 4 mm) [23,24].

The most commonly used materials for the synthetic artificial artifacts are PET-Dacron and PTFE. Dacron is a multifilament polymer, which is formed into artificial vessels with various knitting methods. In the first mode, the fibers are wrapped in a simple pattern, in which they are grouped together and arranged to each other. In the second mode, fiber sets are closely interconnected in the whales. Such structures are relatively strong and almost unmanageable, strongly reducing the likelihood extension and stretching after integration. The tightness between the fibers and, consequently, the porosity of the material can vary during the production process [25]. Due to high permeability, plaited artificial vessels are impregnated with albumin or collagen [26,27]. The treated surfaces of the veins were further improved by chemical treatment with glutaraldehyde, formaldehyde, polyethylene glycol or heparin [28]. Despite the various properties, the artificial vessels are still not optimal. Dacron and polytetrafluoroethylene cores have many positive properties, but smaller diameters are still problematic. High hydrophobicity of the surface limits the endothelization of the surface. After integration in the organism, infections continue to occur. The greatest problem is the thrombogenicity of the surface, as in many cases patients experience thrombosis, which continues to occur [29,30].

### 2.2. Biocompatibility of Synthetic Materials

The use of synthetic materials in medicine has been growing steadily since the 1940s when they were actually applied in practice. Millions of such products are used each year. Despite all the advances and more than 50 years of research in this field, we still did not create a material that would meet all the requirements and, after application in the human body, be completely without any negative response—in this case materials would be completely biocompatible. Biocompatibility is defined as the ability of a material to induce an appropriate response in a specific application in host [13]. Hemolytic, toxicological and immune responses in the case of materials that come into contact with blood are no longer as problematic. For these materials, the main problem is thrombogenic reactions and the possibility of bleeding after implantation. There are many examples of clinical complications of cardiovascular devices. Thus, complete blockage of stents is reported already within a few weeks after implantation, acute thromboses in vessels of middle-diameter, embolisms in catheters and heart valves, complications of coronary artery bypass, etc. [4]. These problems occur despite therapy with drugs that prevent blood coagulation and the formation of clots. There are more hypotheses as to why despite all the knowledge and the long-term effort, we still do not have a fully compatible surface. One says that it is not possible to produce the industrial surface, which has the same characteristics as the natural one. Natural blood vessels have a layer of endothelial cells which is constantly renewed and thus produces antithrombotic substances, such as, for example, prostacyclin. It is produced in response to the conditions in the blood and is constantly changing. Another antithrombotic substance produced is glycocalyx molecules, which are in contact with the vascular endothelial cells of the blood vessels and due to their composition represent the antithrombotic surface. There are many attempts to imitate the natural blood vessel condition as closely as possible, but for now none are successful. Another hypothesis concerns the knowledge of blood and the processes that are associated with platelet activation and coagulation, but this is also very complex and still not fully understood. Platelets’ membranes contain over 100 different oligosaccharide and protein receptors, which are important for transmitting signals between environmental factors and platelets. It is known that when the synthetic material is in contact with blood, it first comes in contact with the protein and the formation of a protein film [31]. Recently, a lot of attention is paid to this subject, but there are controversial opinions in the literature about whether platelet activation depends more on the amount of blood plasma protein adsorbed on the surface or on the final layer of the protein conformation [32]. Another problem is that there are no standardized methods that could determine the biocompatibility of materials. Research is mostly performed on individual blood components, under different conditions, which are difficult to compare with each other. For all the ongoing research in this field, there are many materials that are potentially better than those used up to now, but there is still a long way to their application, as they have to be tested and obtain the necessary documentation, which is consequently connected with high costs [33,34,35,36].

### 2.3. Factors That Influence the Biocompatibility of Biomaterials

The biocompatibility of the material is largely influenced by the surface properties of the material. The first few atomic layers of the material surface present the biointerface between the cells and biomaterials. Surface characteristics also trigger biological response after contact with the tissue and, ultimately, the success of the transplant or a medical device made of such material depends on it [37]. There are numerous conditions, which overlap and determine the biocompatibility. These are not only the mechanical and chemical characteristics of the material, but also place of application, individual host reaction, immune system as well as physical condition of the patient. According to Ikada, chemical and physical characteristics of the surface, which are responsible for biological response at the interface, are of the greatest importance. In the literature, there are various opinions about which surface properties are crucial for optimal biological response [38,39,40,41]. The most frequently investigated properties are the chemical composition of the surface, the topography and the wettability of the surface. The known effects on hemocompatibility as well as on cell response are presented below.

#### 2.3.1. Impact of Plasma Treatment on the Hemo- and Biocompatibility of Synthetic Materials

Plasma modifies the surface morphology and increases surface roughness of PET. It was shown that such surface modification has a significant effect on platelet adhesion and activation. Even a short exposure of PET surface to highly non-equilibrium plasma reduced adhesion and activation of platelets mainly through oxygen surface functionalization. However effects of plasma treatment diminish with time and many oxygen functional groups are lost from the surface within 3 h of aging [42]. Plasma treatment also has an influence on the biological response, as all plasma treated surfaces exhibit improved proliferation of fibroblast and endothelia cells. The number of adherent platelets practically did not change after nitrogen plasma treatment, however, a much lower number of adherent platelets was observed on oxygen plasma treated surfaces [43]. Cvelbar et al. [44] studied the fabrication of micro- and nanostructure poly(ethylene terephthalate) (PET) polymer surfaces used for synthetic vascular grafts and their hemocompatible response to plasma-treated surfaces. The surface modification of PET polymer was performed using radio frequency (RF) weakly ionized and highly dissociated oxygen or nitrogen plasma to enable the improved proliferation of endothelial cells. Results indicate that surface treatment with both oxygen and nitrogen plasma improved the proliferation of endothelial cells, which increased with treatment time by 15 to 30%. This phenomenon was explained by the creation of new functional groups and the modification of surface morphology, which promotes the adhesion of endothelial cells. Numerous studies have proved that plasma treatment significantly improves biocompatible properties of polymer materials [17,45,46,47,48,49,50]. In their study, Jaganjac et al. [49] proved that oxygen rich coating after plasma treatment promotes binding of proteins and endothelialization of polyethylene terephthalate polymer. In another study it was shown that cells prefer to adhere on moderate hydrophobic polymer surfaces, rather than on hydrophilic or super-hydrophilic ones. Recek et al. [47] showed improved proliferation on oxygen plasma treated polystyrene. On the other hand, Garcia et al. [40] described greatly improved cell proliferation of HaCaT keratinocytes on collagen films modified by argon plasma treatment. There are countless papers in the literature describing the improvement of hemo- and biocompatibility of synthetic polymer materials using plasma treatment. Only a few were presented in this paragraph, proving that plasma really is a good method to improve polymer properties for biomedical applications.

#### 2.3.2. The Effect of the Surface Chemical Composition on the Hemocompatibility of Biomaterials and on Cell Response

The chemical composition of the surface is one of the key characteristics when designing the hemocompatible materials from which medical devices are made. There are various ways in which we can control the surface with specific chemistry and functional groups, such as, for example, hydroxyl, methyl, sulphate, carboxyl, amino group etc. [51,52]. Their purpose is to improve the immobilization of various biomolecules such as proteins, enzymes and so on to improve the cellular response. Cell interaction with the surface of biomaterial is never direct, because the surface is previously covered with water molecules and proteins absorbed from biological fluids (see Figure 1). Initially cells respond to this adsorbed protein layer, rather than to the surface itself [53]. Cell adhesion is conducted in several phases: an early phase where short-term events take place, like physico-chemical linkage between cells and material, and a later signal transduction phase, involving biomolecules like extracellular matrix (ECM) proteins, cell membrane and cell skeleton proteins, regulating the gene expression [54]. These phases are illustrated in Figure 1. Firstly, when the biomaterial is in contact with cells in vitro or when they come in contact with an implant surface in vivo, the proteins either from culture medium or biological fluids adsorb and form the protein layer on the surface. After that, cells attach on the surface covered by proteins, spread and express cytoskeleton proteins and integrins, which help them firmly adhere to the surface. Thirdly, the proteins connect, and cytoskeleton reorganizes to adapt the surface morphology and actively spread on the substrate. Finally, at the interface with the material, cells synthesize ECM proteins, securing their shape stability and cell-matrix-substrate interfaces [55,56]. Cell adhesion differs on cell phenotype, that is why mechanisms of adhesion of blood cells are different from mechanism of cells from connective tissues, like fibroblast, osteoblasts, or cells originated from endothelia and epithelia, like endothelial vascular cell or keratinocytes. Cells from connective tissues use mostly integrins in cell-ECM interactions, whereas epithelial and endothelial cells can adhere with both adhesion molecules.

The results of many studies have not yet led to the solution of what an ideal surface should be. Grunkemeier et al. [12] reported that increased oxygen groups reduced the activation of coagulation. Likewise, a higher proportion of these groups should also affect the reduction in the amount of bound fibrinogen and, according to their results, also the reduction in the number of bound platelets [57]. The same authors also reported that coagulation was reduced when methyl groups on the surface increased. Tengvall and colleagues [58] came to the same conclusions. One way to introduce new functional groups on the surface is plasma treatment, where different surface functionalities can be achieved with different types of plasma. Wang et al. [59] treated the PET polymer surface with acetylene plasma, thereby increasing the carbon content of the surface. Such films at different atomic percentages acted inhibitory at adhesion and activation of platelets. A significant reduction in contact activation of platelets was observed in the treatment of polyurethane with nitric plasma [60]. On the other hand, improvement was not observed in hemocompatibility after treatment with oxygen and argon plasma. Better cell adhesion and proliferation was observed on oxygen plasma treated PET and polystyrene (PS) samples, while samples treated in CF_4_ or nitrogen plasma did not show significant improvement [45,46,47,48]. Jaganjac et al. [49] found that oxygen functional groups on PET treated by plasma, stimulated endothelial cell growth and proliferation by 25%, compared to control, plasma untreated samples, suggesting the possible use of oxygen plasma treatment to enhance endothelialization of synthetic vascular grafts.

#### 2.3.3. The Influence of the Topography on the Hemocompatibility of Biomaterials and on Cell Response

In the production of biomaterials, it is important to take into account the impact of the structure of the material on the biological response. Natural vessels have of course the best biocompatibility, so it is necessary to know their structure, which consists of several layers. The first layer inside the vessel consists of endothelial cells that connect with the basal lamina. Its main building blocks are collagen, proteoglycans and glycoproteins such as fibronectin and laminin. They follow the layers of elastic fibers and smooth muscle cells that shrink under the control of a sympathetic nervous system. The back or the outer layer builds connective tissue. The surface of the inner side of the vessel is not smooth, but it is made up of micrometer corrugated grooves running in the direction of the blood stream. At the top of the individual protrusions, there are nano projections. Significance of the influence of the topography on biological response has been brought to the attention of many researchers, all seeking to create structures that will achieve the highest degree of hemocompatibility. There are many physical and chemical methods or combinations of both to achieve nanostructure, either by deposition of the material or by etching it. The most commonly used methods are: photolithography, colloidal lithography, laser etching, metal oxidation, nanophase ceramics production, supramolecular aggregation, surface coating with carbon nanotubes, nanowires, nanocomposites and, last but not least, plasma techniques such as plasma chemical infiltration, ion implantation, plasma sputtering, etc. [61,62,63]. Fan and co-workers [64] created structures with grooves of about 500 nm in width and about 100 nm in height, and 100 nm × 4 nm in size on the surface of polydimethylsiloxane (PDMS) using self-assembled layers and lithography. For comparison, unmodified, smooth PDMS surface, surface with ditches and surface with only nano-extensions were used as control. Platelet adhesion analyses have shown that only on the surface containing both grooves and protrusions was the number of bound platelets significantly reduced. Until recently, it was established that the increase in roughness of the surface due to the greater surface area available for platelet binding also increases the level of thrombogenicity. In the literature, the most commonly used term for characterization of the surface is roughness, but it is important to be aware of the fact that it does not tell us much about the actual topography, but only gives the average roughness of the surface. To explain the observed hemostatic response, we proposed few hypotheses in this review. One of the hypotheses says that hemostatic response is based on significant reduction of contact area between polymer and platelets due to high roughness of plasma treated polymer samples. According to Chen and colleagues [62], the roughness values that are crucial for platelet binding are roughly divided into three groups. The first group includes areas with a roughness of more than 2 μm, which is about the size of the platelets. The second group are surfaces with a roughness less than 2 μm, where the correct design of the structures can reduce the contact area of platelets and, consequently, the platelet adhesion, since they can be fixed only at the top of the structures as shown in Figure 2.

The third group includes roughness greater than 50 nm, where the surface structures are much smaller than pseudopods. These are smooth surfaces that do not play a role in platelets adhesion. In such cases, other factors are likely to be involved in reducing thrombogenicity, among which are binding and conformation of plasma proteins, in particular fibrinogen and albumin. According to other authors, when performing structuring of the surface is important to take into account the entire dimension of the surface (height, width and distances between individual structures) [65,66,67]. It was shown in many studies that cells respond to surface topography and align themselves along defined surface features, e.g. ridges or grooves [68]. However, cell behavior on the nano-topography is still unknown so far. In the review of Curtis and Wilkinson on topological control of cells, they correlated the topological parameters with biological parameters, such as short- and long term adhesion and proliferation [69]. In several studies [69,70,71] it was demonstrated that the best adhesion of human bone cells was on less organized rough surfaces. Dalby et al. [71] developed surfaces with 120 nm diameter nanopits and demonstrated stimulation of human mesenchymal stem cells on such surfaces. In 1997, Curtis and Wilkinskon [68] described that cells reacted to discontinuities on the surface, with a radius of a certain length. It is related to the cell mechanism of mechano-sensitivity which is related to integrin mediated cell-matrix adhesion [72]. There have been different hypotheses on how cells sense the morphological discontinuities on the surface. One hypothesis is about the thermodynamics and extra-cellular matrix protein adsorption [73], another is about discontinuities acting as the energy barrier, where the size of energy barrier depends on both the geometry and surface chemistry [74]. Stevenson and Donald [75] have been investigating the attachment of cells on the different micro-meter scale substrates. They observed that the attachment of cells is dependent on the ridge spacing. At the ridge spacing between ~10–20 μm, the cells were able to attach and bridge between the neighbor ridges. At the moderate spacing, from ~30–50 μm cells attach to a single ridge or groove and at the largest spacing ≥50 μm cells connect between a ridge and a groove. From these results they proposed both a critical length and a critical slope angle of the ridge-groove surface morphology. Cells adjust their shape according to morphology, which causes reorganization of attachment and cytoskeleton structures (see Figure 3). Similarly, Berry et al. [76] described that cells were sensitive to the changed morphology, especially in the radius of curvature of pits. Thery et al. [77] have noticed that cells can memorize and recognize the adhesive substrates and in this way reorganize attachment and cytoskeleton structures. Hallab at al. [78] demonstrated that for cellular adhesion and proliferation, even more important factor is surface free energy of polymers. Other groups demonstrated that short term cells adhesion on metal substrates coated with gold-palladium is more dependent on surface chemistry, whereas the long-term adhesion is more dependent on surface roughness [70,79,80,81,82]. Ponsonnet at al. [83,84] also observed high impact of surface energy of titanium and titanium alloys on cell proliferation.

#### 2.3.4. The Effect of Wettability on the Hemocompatibility of Biomaterials and on Cell Response

Wettability is one of the important properties of the surface and has a major impact on the biological response. The second hypothesis is explained below and is based on preferential adhesion of water molecules from blood to the polar functional groups on the polymer surface. When we talk about the wettability of the surface, this is most often associated with the adsorption of proteins. In general, hydrophobic surfaces are considered to be much more susceptible to protein binding than hydrophilic, due to the strong hydrophobic reactions resulting from the contact of the protein with the surface, which results in reflective forces due to strongly bound water molecules. In addition to the amount of bound proteins, wettability also affects the conformation of bound proteins. Because contact of artificial material with blood leads to immediate contact with proteins, this is consequently important for the binding and activation of platelets and hence for the hemocompatibility of the material. Blood cells adhere with membrane adhesion proteins cadherins and selectines, which are involved in cell-cell interactions, and integrins which are involved in cell-material interaction [85,86,87]. 

Xu and Siedlecki [88] treated polyethylene with gaseous plasma and created different wettability of the surfaces. The influence of such surfaces on the binding of proteins was checked by binding of three blood plasma proteins. For all three proteins, fibrinogen, bovine serum albumin, and Human Factor XII, critical values were found to be at the water drop between 60° and 65°, where adhesion increased at these values. By measuring the force with AFM tip, it was also found that the binding forces change over time, which suggests that after binding of proteins to the surface, their conformational changes occur. Similar results were also obtained on the polyurethane polymer, where the increased binding of fibrinogen was also observed at angles greater than 65°. The conformation was monitored by the binding of monoclonal antibodies, and it was found that it varies with different surface wettability, depending on the ability of binding antibodies to different binding sites on the fibrinogen molecule [89]. The binding of molecules to surfaces should be a time-dependent process, which is supposed to be carried out on several levels. The first molecules which come into contact with the surface are water molecules, which also react with the surface according to its properties. Water molecules create a layer on the surface from which the binding of other molecules depends and diffuse later because of their size to the surface. If there is a mixture of different proteins in the solution, their binding depends on both size and their properties. Over time, their exchange can occur, as dynamic confocal changes and reorientation can affect the binding power and consequently the activity of the protein [66,88]. Due to the redistribution of amino acids, the availability of receptor binding sites may also change, which could also affect platelet adhesion. The influence of surface wettability on platelet binding is also the subject of numerous studies. In 2002, Spijker and colleagues [90] studied adhesion and platelet activation on polyethylene, in which gaseous-plasma produced a gradient of hydrophobicity and concluded that binding was greater on hydrophilic surfaces, and their activation was greater on more hydrophobic surfaces. Vogler and colleagues [91] in 1995 came to the same conclusion. Rodrigues and co-workers [57], Lee et al. [92] and Sperling and colleagues [52] came to the exact opposite conclusion. They listed the largest number of both bounded and activated forms of platelets on hydrophilic surfaces. Cell attachment was investigated by Yanagisawa and Wakamatsu [93]. They observed that cell attachment rate and cell spreading were higher on substrates with a water contact angle below 60° and that attachment decreased dramatically for more hydrophobic surfaces, whatever the time after inoculation. For the oblast cells, no correlation between wettability of the material and cell attachment and proliferation was found [94]. On the other hand, Lee et al. demonstrated that endothelial cells [95] or neural cells [96] adhesion was more increased on moderately hydrophilic surfaces, than on the superhydrophilic or hydrophobic surfaces. In 2004, it was demonstrated by Lime et al. [97] that hydrophilic substrates are better for human fetal osteoblast adhesion and proliferation than hydrophobic ones. Interesting, surface energy had no effect on cell differentiation.

Despite the fact that the wettability of the surface is likely to play an important role in the hemocompatibility of materials, it is difficult to derive clear conclusions from the contradictory results, which could explain the role of wettability. One of the reasons is definitely the complexity of the processes that take place in the blood. Individual impacts cannot be considered separately. Such a multivariable system should be taken as a whole and take into account the interaction between the individual impacts.

## 3. Biomaterial-Blood Interactions

When biomaterial comes in contact with the biological system, activation of the intrinsic pathway at the blood/biomaterial interface starts. There are many studies examining blood biocompatibility and the most important parameters for characterization are the number of adhered platelets and their activation [98]. Platelets are the smallest blood fragments with a diameter of 1 μm to 3 μm, without nucleous. In the blood of an adult, they are 2–3 × 10^8^ per mL. In the bloodstream, they are present in inactive form. In the event of endothelial vein damage or when activating the coagulation cascade, the platelet shape is activated and changed. Initially, it was established that platelets are important only in stopping bleeding, but now it is known that in addition to these very important functions, they also play an important role in other physiological and pathological processes of hemostasis, inflammatory reactions, tumor metastases, and defense mechanisms [99].

Figure 4 represents the artificial PET polymer material in contact with blood. The surface of original material (as manufactured at the factory) is fully covered with platelets (Figure 4a). However, when PET is treated with oxygen plasma, there are only few platelets adhered on the surface of material (Figure 4b). When the body is in a contact with artificial biomaterial, platelets tend to adhere similar like in the case of an external injury. This is the reason why materials, which show strong platelet adhesion or provoke an increase in platelet adhesion, are considered as thrombogenic [100]. If the blood leucocytes decrease at the same time, this is a sign of a “cellular immunoresponse” of the body towards the biomaterial. Material is considered as clinically biocompatible when it does not provoke any damage of blood cells or any structural change of plasma proteins when in contact with the blood [101]. Human blood plasma contains over 300 different proteins that differ in structure and function: proteins involved in coagulation and fibrinolysis, complementary system proteins, immune system proteins, enzymes, inhibitors, lipoproteins, hormones, cytokines and growth factors, proteins that are important for transport and others [102].

To explain hemostatic response, third hypothesis is based on different conformations and orientations of adsorbed plasma proteins. If proteins adsorb on the surface and blood cells adhere on the surface of material, the contact of the biomaterial with blood leads to clot formation [103,104,105]. Activation of the coagulation system at the blood-biomaterial interface drives sequence of reactions. Proteins compete to adhere to the biomaterial surface and this determines the pathway and adhesion of platelets. Having the exact knowledge of the material surface and the conformation of the adsorbed proteins, prediction about the interactions between the biomaterial surface and the absorbed proteins can be made. These interactions are determined both by the nature of the polymer surface and by the nature of protein parts in contact with the surface (hydrophilic/hydrophobic, charged/uncharged, polar/non-polar etc.) [106,107,108]. It is commonly accepted that a decrease in surface roughness increases the compatibility of material [109]. Surface tension of a material is one of the most important factors on protein adsorption. Andrade et al. [110] suggest that smaller interfacial energies between blood and polymer surface results in better blood compatibility. Contrary, Bair et al. [111] claims that higher surface tension, between 20–25 mN/m gives better hemocompatibility. On the other hand, Ratner et al. [112] prove good blood compatibility on the surfaces with the moderate relationship between their hydrophilic/hydrophobic properties. Carboxylate, sulfate or sulfonate groups on the surface may act as antithrombotic agents, as a result of repulsive electric forces between plasma proteins and platelets [113]. Norde has shown that protein adsorption increases if concentration of ionic groups in the protein and in the polymer surface decreases [114]. The relation between the electrical conductivity of biomaterials and blood biocompatibility is described by Bruck [115]. In addition, there are studies on the influence of the streaming potential on blood coagulation [114,116]. 

Hemostasis is the body’s response to vascular damage and bleeding. It involves a complex set of events and biochemical reactions that lead to the formation of a blood clot which consequently prevents bleeding. At the beginning of the 20th century, Morawitz combined all the insights into his classic coagulation theory, which he divided into two groups. In the first, in the presence of calcium ions and thrombokinase, the conversion of prothrombin into thrombin occurs. In the second step, the resulting thrombin converts fibrinogen into fibrin. His theory touched the basics of coagulation, but it had several drawbacks. One of them was that it did not take into account the specific function of platelets, which was later described by Bürker. Fibrinogen is a large, complex bar shaped glycoprotein. It consists of three pairs of Aα, Bβ and γ polypeptide chains, which are interconnected with 29 disulfide bonds. At both ends, globular domains are interconnected with α-helices and bind calcium ions, which are important for maintaining the structure and function of fibrinogen. In blood plasma, it is usually present at a concentration of about 2.5 g/L. Fibrinogen is important for the preservation of hemostasis and platelet aggregation [57,102]. This protein also plays an essential role in binding to synthetic materials and thus has an important impact on the hemocompatibility of the material. In addition, similar to protein fibrinogen, albumin is also important in binding to the surfaces of synthetic materials, helping to maintain the surface antithrombotic [102]. Human serum albumin is the most abundant protein in human blood plasma. It is synthesized in the liver and is present in all body fluids. It consists of a single chain containing three interconnected domains. It has binding sites for various molecules like water, ions, fatty acids, hormones, bilirubin, synthetic medicines and many others. It is present in blood plasma at concentrations of 35–50 g/L. Because of its abundance and high binding capacity, albumin is the main transport protein that regulates and maintains osmotic pressure in the blood. Competitive adsorption of the protein albumin and fibrinogen is very complex and has been widely investigated. Albumin inhibits and fibrinogen activates the adhesion of platelets; in the case of hydrophobic surfaces, fibrinogen is mostly absorbed, while in the case of hydrogels, absorption of albumin is dominant [117,118]. The stationary state, which corresponds to an irreversible protein adsorption, is reached after longer contact time. The adsorbed protein films show time-dependent conformational changes, like desorption or protein exchange and are described by the Langmuir isotherms [119,120,121].

Research in this field has led to many different theories and numerous terminologies. Much progress was made when the International Commission introduced a common name for coagulation factors, which have since been designated with Roman numerals [122]. Understanding the processes of coagulation, which is established at the present time, is a result of years of research, but still there are numerous questions waiting for answers [123,124].

## 4. Methods for Improving the Biocompatibility of Synthetic Materials

Nevertheless, their chemical structure, hydrophilicity, roughness, crystallinity and conductivity are not suitable for certain applications and need to be modified [11]. A number of methods are available to improve biocompatibility of biomaterials. The most promising method is coverage of synthetic surfaces with a monolayer of human endothelial cells, since this closely imitates biological conditions. In natural blood vessel, a monocellular film of endothelial cells covers the interior of a vessel, which is in contact with blood and has an important function in blood compatibility [125]. Another common method is chemical surface modification, by including specific functional groups on the surface. These methods are relatively invasive and may also result in harmful chemical products that may lead to irregular surface etching on one hand and may be harmful to the environment on the other. Modification of a material with surface functionalization can also be achieved by ozone oxidation or gamma radiation and UV radiation, but these methods do not achieve a lasting effect, and there is also a high probability of polymer degradation [126]. Most of these disadvantages can be replaced by plasma treatment, which proved to be a very promising method for optimization of surface properties of synthetic materials [127,128,129]. 

### 4.1. Plasma Treatment of Polymers

Plasma treatment is an environmentally friendly method that enables easy and fast modification of the surface of polymers, whereas the polymer bulk properties remain unchanged. Plasma treatment causes formation of new functional groups on the surface, increase of surface energy, increase or decrease of hydrophobicity and hydrophilicity, change of morphology and roughness, and increase or decrease of polymer crystallinity. It also removes poorly bound layers and impurities. The reactions occurring during plasma treatment can be divided into several groups. Surface reactions as a result of plasma changes create functional groups between atoms present in gas and surface atoms and molecules. Such reactions can be achieved with oxygen, nitrogen and NO_2_ plasma. With plasma, thin films from organic monomers, such as CH_4_, C_2_H_6_, C_2_F_4_ to C_3_F_6_, can also be formed. Such polymerizations involve reactions between atoms in gas and on the surface of polymer and reactions between surface molecules. Plasma can produce volatile products from the surface of polymers by chemical reactions or by physical etching, thus removing unwanted material from the surface. Oxygen plasma is used to remove organic impurities such as oligomers, antioxidants, by-products released from molds and other microorganisms. Oxygen and fluorine plasmas are commonly used for etching of polymers [128,130,131]. Oxygen and mixtures of oxygen plasma are also widely used for treating materials, which are used in biomedical applications [47,49,67,129].

The main products generated during treatment in non-thermal plasma are electrons, ions, excited particles, radicals, as well as UV radiation. These products are mainly free radicals, unsaturated organic components, cross-links between polymer macromolecules, degradation products of polymer chains and gas products. The effects of electrons and UV radiation cause the R–H and C–C bonds to break, which can be represented by the following reactions [128]:(1)$$RH\to R^{•}+H,\;RH\to R_{1}^{•}+R_{2}^{•}$$

The direct formation of unsaturated organic compounds with double bonds on the surface of polymers describes the following reaction:(2)$$RH\to R_{1}-CH=CH-R_{2}$$

In the secondary reactions of atomic hydrogen through various mechanisms, molecular hydrogen is usually formed, including recombination and transfer of hydrogen to polymeric molecules. These reactions describe the following equation:(3)$$H+H\to H_{2},\;H+RH\to R^{•}+H_{2}$$

In addition to recombination, in organic material atomic hydrogen may also form a double bond with an organic radical:(4)$$H+R^{•}\to R_{1}-CH=CH-R_{2}$$

During the treatment of polymers with non-thermal oxygen plasma, free organic radicals form on the surface and react with molecular oxygen in the gas phase form, resulting in the formation of active peroxide radicals [128,132,133]. This process describes the following equation:(5)$$R^{•}+O_{2}\to R-O-O$$

These RO_2_ peroxide radicals can trigger various other chemical reactions. The simplest processes involving RO_2_ radicals are reactions where various peroxide components are formed on the surface of the polymer and can be simplified by Equations (6) and (7). Due to the low energy of electrons and ions in plasma and the high excitation coefficient of UV radiation, these peroxides on the surface are formed in a thin layer [134,135].
(6)$$R-O-O^{-}+RH\to R-O-O-H+R^{•}$$
(7)$$R-O-O^{-}+RH\to R-O-O-R_{1}+R_{2}^{•}$$

In addition to the formation of new functional groups on the surface of polymers, the plasma treatment also produces an effect called etching. Etching can be explained by two mechanisms: chemical etching and physical etching, which occurs due to ion bombardment. Chemical etching results in surface reactions, which makes the surface part of the polymers to fumigate. The major molecules that usually participate in these reactions are oxygen atoms, ozone, fluorine atoms and electronically excited oxygen molecules. Both processes take place during plasma treatment, so it is difficult to separate them from each other. Nevertheless, by changing the conditions and characteristics of plasma, we can regulate the relationship between the two processes. In addition to the processing conditions, the polymer etching rate depends also on the type of gas used. In the case of PET polymer treated with different plasmas at different power levels (25, 50 and 100 W) and frequency of 13.56 MHz, it was found that the etching was linearly dependent on the power [136]. The highest degree of etching occurred in oxygen plasma. In addition, etching also depends on the type of polymer, its chemical composition and the crystallinity of the material [135].

#### 4.1.1. Aging of Plasma-Treated Materials

The stability of the plasma treated surface is a very important feature, especially if materials are not used immediately after plasma treatment. Plasma treated biomaterials, used in medicine, where materials come in contact with the living tissue, stability is very important. After plasma treatment, the surface of polymers tends to return to its original state, what is called ageing. Many researchers studied the so-called aging of various materials that were treated with different types of plasmas. Aging depends on both the type of plasma and the treated material. Experiments on polydimethylsiloxane, which was treated in nitrogen, oxygen, argon and NH_3_ plasmas and aged on air and in the buffer solution confirmed, that the surface returned back to initial, hydrophobic state after one month [137]. Wilson and colleagues [60] treated the surface of PTFE polymers with the same types of plasma and were aged under the same conditions. From their studies it was concluded that aging was present under both storage conditions and that the effect was more noticeable in aging in the buffer. For both samples, aging after one month stabilized, but the condition did not return to the initial state. On the other hand, the hydrophobicity of the polysulphonic membranes remained unchanged even after three months after treatment with CO_2_ plasma [138]. Modic et al. [42] studied aging of PET polymer treated in oxygen plasma and exposed to different environmental conditions (see Figure 5). After plasma treatment, the contact angle dropped from original 73° to 10°. The first set of treated samples was left at room temperature; the second set was stored in the refrigerator at 4 °C and the third set was put in phosphate buffer solution (PBS). Aging of all samples were monitored for two weeks. Results showed that samples stored at room temperature and those stored in the refrigerator had the same relative slow aging; the contact angle changed from original 21° to 30°; again ageing in PBS turned out to be much faster. Already after 3 h contact angle increased from 10° after plasma treatment, to ~30°. The effect of ageing in all environmental conditions was observed for the first three days; later on the contact angle does not change significantly (Figure 5).

There are four proposed mechanisms of ageing:Reorientation and relocation of polar groups from the surface of the polymer into the bulk of the material due to thermodynamic relaxation,Diffusion of low molecular weight oligomers from the interior to the surface and products that are formed during plasma treatment on the surface of polymers,Reactions of free radicals and other active species and groups formed during treatment, with each other and with the environment in which the polymer is located.

The aging of hydrocarbon materials treated in oxygen plasma is mainly due to the reorientation and transfer of polar peroxide groups into the interior of the polymer. If the same polymers are treated with nitrogen plasma, aging results from reactions of nitrogenous surface groups with the environment after plasma treatment [135]. 

## 5. Conclusions and Future Perspectives

Plasma treatment is one of the most favorable methods for treatment of synthetic materials and it greatly improves the hemocompatible properties of polymers. Systematic measurements on whole human blood of healthy volunteers have confirmed the hypothesis that the rate of hemocompatibility monotonically increases with increasing hydrophilicity and surface roughness. Biocompatibility depends on the success rate of surface endothealization, which is strongly correlated with surface properties, i.e., surface wettability, topography and chemistry. According to studies, cells prefer to adhere to moderate hydrophilic surfaces at micrometer scale. Furthermore, oxygen functional groups on the surface proved to stimulate cell adhesion and proliferation.

Based on extensive experimental results, three possible hypotheses to explain the observed hemostatic response were proposed in this review. The first hypothesis is based on preferential adhesion of water molecules from blood to the polar functional groups on the polymer surface. The second one is based on different conformations and orientations of adsorbed plasma proteins, and the third hypothesis is based on the significant reduction of contact area between polymer and platelets due to high roughness of plasma treated polymer samples. Because of extreme complexity of interactions between whole blood and polymer surface, it is not possible to declare which hypothesis is the most suitable. Results indicate that a combination of different physical and chemical processes can lead to a biological response of material used for fabrication of artificial grafts and other cardiovascular implants. 

## Figures and Tables

**Figure 1 materials-12-00240-f001:**
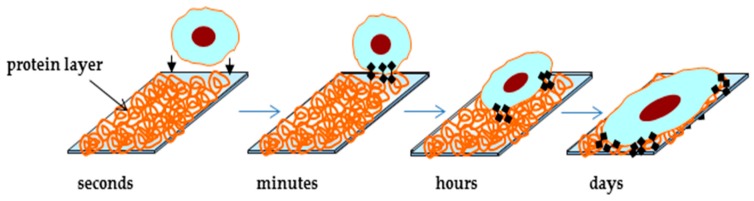
Kinetics and phases of cell adhesion.

**Figure 2 materials-12-00240-f002:**
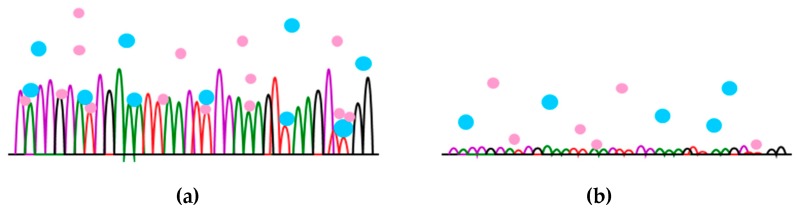
Nanostructured topography of the surface, that attracts (**a**) and repels (**b**) adhesion of platelets.

**Figure 3 materials-12-00240-f003:**
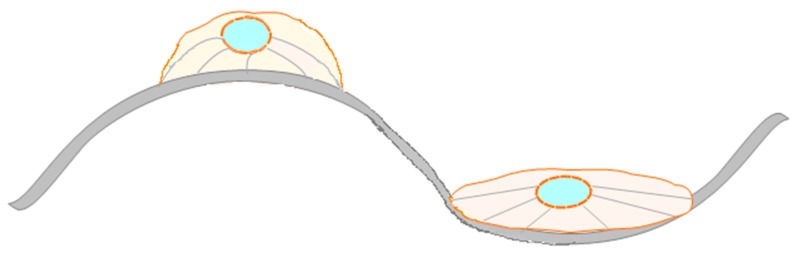
Illustration of the reorganization of cell actin skeleton structures, shape and attachment according to surface morphology.

**Figure 4 materials-12-00240-f004:**
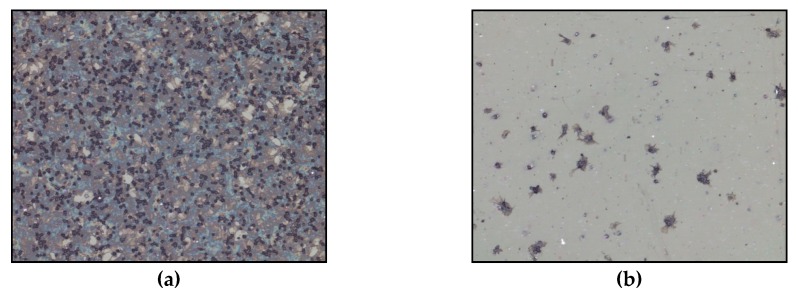
Platelet adhesion on untreated (**a**) and oxygen plasma treated (**b**) poly(ethylene terephthalate (PET) polymer. Incubation was performed with shaking at 250 RPM.

**Figure 5 materials-12-00240-f005:**
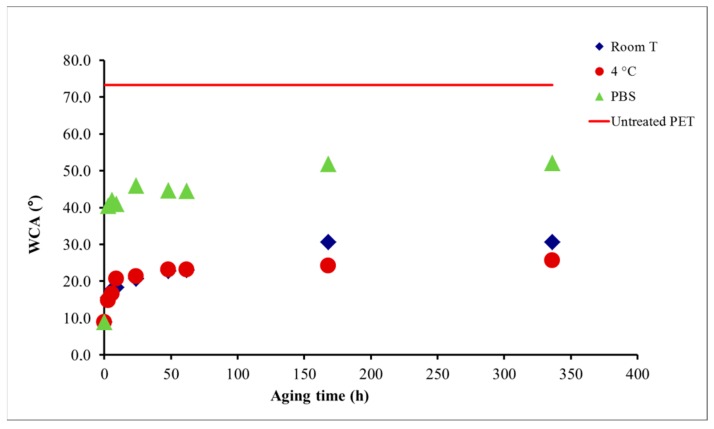
Effect of different aging conditions on the wettability of surface of PET polymer treated in oxygen plasma glow for 30 s.

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
