# Peer review of "Biocompatibility of Plasma-Treated Polymeric Implants"

_materials, 2019, doi:10.3390/ma12020240_

Reviewer 1 Report

The manuscript is well structured and the topic falls into the MATERIALS aim and scope. however some minor points should be addressed by the author to have a more complete paper. 

The introduction is weak. Some more references should be cited in order to better present the rationale of the study underlinying aim and scope. The primary and secondary outcomes are missing as well as the clinical possible implication of the study.

Moreover it is unclear the order of the paragraphs. Introduction is paragraph 1 then there is a paragraph number 3 named "biomaterials".... paragraph number 2 is missing. 

Author should better organize the paper accordingly with the Journal aim and scope.  

Author Response

please find our response in attachment.

Reviewer 2 Report

I have read with interest the review by N Reek entitled: “Biocompatibility of Plasma-Treated Polymeric Implants “

On the form, the manuscript is well written and is rich of information and references regarding biomaterials interaction with the body. Nevertheless, there is an inconsistency between the title and the abstract of the manuscript from one side and the actual content of the manuscript.

1-  The manuscript seems to be a segregation of different notions that are treated in paralleled without a conductive line:

a.      author starts by describing the biomaterials.

b.      Then there is a paragraph about artificial vascular graft.

c.      After that the author goes in details through biocompatibility of synthetic biomaterials and the factors that influence it.

d.      Next, she enumerates methods that are used to improve the biocompatibility.

e.      At the end of the manuscript there is a paragraph about the plasma treatment of polymers in which the author described the chemistry of the plasma action o the surface, and continues with a paragraph in which she explains the problem of aging of plasma treated surfaces.

f.       Finally, the author gives general information about cell adhesion to the surface of a materials.

2- Aside from very few individual examples dissimilated in different paragraphs, there is not a paragraph dedicated to review the impact of plasma treatment on the biocompatibility, which is the object of the review according to the title and the abstract. 

3- The conclusion starts with the following statement: “Plasma treatment is one of the most favorable methods for treatment of synthetic materials and greatly improves the hemocompatible properties of polymers”.

In fact, within the manuscript there is no paragraph dedicated to the presentation of literature data about the improvement of hemocompatibility using plasma treatment.

 4- In the abstract the author state: “several hypotheses to explain the improvement in hemocompatibility of plasma treated polymer surfaces were proposed”.

There is no such hypothesis in the manuscript.

5-In paragraph “6. Methods for improving the biocompatibility of synthetic materials”, Author States that “chemical surface modification can lead to adverse effects such as loss of mechanical properties and faster degradation processes. In addition, it is difficult to control the mechanisms, resulting in poor reproducibility of the results”.

I completely disagree with these critics: there is no evidence in the literature that chemical modification of the surface impacts mechanical properties of the bulk materials and these methods of chemically surface modification are very reproducible both at laboratory and industrial level.

6 – some paragraphs present very basic information’s that are not useful at all for the object of the review, it’s the case for paragraph: “5. Blood and blood plasma” and paragraph “7. Cell adhesion”

Minor points:

-        Line 10: “the success of the treatment significantly dependents on the stage of disease progression”.Dependents should be replaced y depends

-        Line 95: “Poor properties are fragility, increased fragility” Fragility is repeated.

-        Line 290: “Latter was confirmed by many authors in their publications.” The phrase is not understood.

-        Line 379: “The main tasks of the blood are follows: Follows should be replaced by “as follows” or “the following”.

For all the above reasons I advise the author to review the context of the manuscript and either transform it into a general review about “biocompatibility and surface impact”, or to enrich the content with plasma treatment data in relation with the biocompatibility in order to present it with the actual title and abstract.

Author Response

please find our response in attachment.

Round  2

Reviewer 1 Report

Author made excellent job addressing all the reviewer note and request

Reviewer 2 Report

The author addressed the main points I revealed in my previous comments. I can recommend the publication of the manuscript in its revised version.

Note: Figure 1 is repeated twice in the revised manuscript (Page 6 and Page 14) and its second copy should be deleted from page 14.